# Stem Cell Therapy for Myocardial Infarction Recovery: Advances, Challenges, and Future Directions

**DOI:** 10.3390/biomedicines13051209

**Published:** 2025-05-16

**Authors:** Nicholas T. Le, Matthew W. Dunleavy, William Zhou, Sumrithbir S. Bhatia, Rebecca D. Kumar, Suyin T. Woo, Gonzalo Ramirez-Pulido, Kaushik S. Ramakrishnan, Ahmed H. El-Hashash

**Affiliations:** 1Department of Biology, College Station Campus, Texas A&M University, College Station, TX 77840, USA; matthewdunleavy@tamu.edu (M.W.D.); summi.bhatia@tamu.edu (S.S.B.); rebeccakumar@tamu.edu (R.D.K.); wootae@tamu.edu (S.T.W.); ksr1234@tamu.edu (K.S.R.); 2Department of Health Promotion and Behavioral Sciences, University of Texas at Austin, Austin, TX 78712, USA; parakeet@utexas.edu; 3Department of Biology and Biochemistry, University of Houston, Houston, TX 77004, USA; gramir45@cougarnet.uh.edu; 4Department of Biology, Elizabeth City State University Campus of the University of North Carolina, Elizabeth City, NC 27909, USA

**Keywords:** myocardial infarction, stem cell therapy, cardiac regeneration, regenerative medicine, cell-based therapy, tissue engineering

## Abstract

Myocardial infarction (MI) is a leading cause of morbidity worldwide, resulting from ischemic damage and necrosis to cardiomyocytes. While the standard treatment regimen for MI can be successful in restoring coronary perfusion, it typically does not resolve myocardial damage, which can leave patients particularly vulnerable to complications such as heart failure or electrical conduction abnormalities. Stem cell therapies offer a promising novel approach aimed at restoring cardiac function and decreasing the incidence of functional complications after an MI. This review used a literature search to evaluate the current landscape of stem cell therapy for post-MI recovery and focuses on the stem cell candidates for MI recovery therapy, delivery methods of such treatment, and their effectiveness. Both preclinical and clinical trials have demonstrated the safety of stem cells, but have struggled with limited cell retention, inconsistent efficacy, and survival. Mechanisms are employed by stem cells to promote regeneration, such as paracrine signaling, angiogenesis, and structural remodeling, in addition to the various stem cell delivery methods, including intracoronary infusion, direct myocardial injection, and intravenous administration. Furthermore, some strategies to combat past challenges in this field are discussed; for instance, extracellular vesicles, bioengineered patches, hydrogels, gene editing, and bioprinting. This article will provide a framework for future research in stem cell therapies and highlight the current progress in the field.

## 1. Introduction

Globally, ischemic coronary artery disease is a leading cause of death. Coronary artery disease frequently deteriorates to myocardial infarction (MI), wherein an occlusion of a coronary artery results in local ischemia causing extensive necrosis in the muscular myocardium, adverse ventricular remodeling, and fibrotic scar formation [1]. Currently, reperfusion therapy is the standard of care for acute myocardial infarction, which consists of percutaneous coronary interventions, pharmaceutical thrombolytic therapy, and coronary artery bypass grafting. These interventions are all focused on swiftly restoring coronary circulation and preventing further tissue damage. However, this approach fails to foster longer-term regeneration of the recently damaged tissue, and complications in the years following a myocardial infarction still constitute a major source of mortality for these patients [2]. Mortality after MI highlights the pressing need for treatment modalities that can promote cardiac regeneration and approximate or completely restore pre-MI cardiac function.

Stem cell-based therapies including mesenchymal stem cells (MSCs), cardiac stem cells (CSCs), induced pluripotent stem cells (iPSCs), and bone marrow derived mononuclear cells (BMMCs) have been at the forefront of regenerative medicine in the treatment of many diseases [3], and have an acceptable safety profile, but their efficacy in cardiac regeneration is a topic of heated debate, as cardiac cells are fully differentiated and upon damage are replaced by fibroblasts. Introduced stem cells, on the other hand, have the ability to differentiate into new cardiomyocytes [4], and can induce paracrine signaling pathways, angiogenesis, and immunomodulation [5,6]. Clinical and preclinical studies have demonstrated the safety of stem cells in addition to increases in left ventricular ejection fraction (LVEF) and reductions in scar tissue proliferation of infarct areas, although some studies have yet to be reproduced, likely due to difficulty with lack of stem cell retainment and survival [7].

Herein, this paper elaborates on the use of various stem cell therapies to induce cardiac regeneration and repair after the damage caused by MIs. There are many newly elucidated strategies that can help address the current issues with stem cells in regenerative cardiology, such as extracellular vesicles, hydrogels, and special patches that allow stem cells to adhere to their transplanted environment [8]. These methods, even if currently not particularly effective, still hold significant potential and will likely eventually be used in the definitive treatment of tissue damage after MI. This article will provide a framework for future research in stem cell therapies and highlight the current progress in the field.

## 2. Literature Review Methodology

A PubMed search was conducted to find articles relevant to the current landscape of stem cell therapy for post-myocardial infarction recovery using the following query: (((“myocardial infarction” [All Fields]) AND (“stem cells” [All Fields])) OR (“cardiac stem cells” [All Fields])) AND (“myocardial infarction” [All Fields]), applying filters for various study types, including clinical trials, comparative studies, experimental studies, and validation studies. The first 100 articles were screened based on recency and relevance to primary focus areas—namely stem cell type, delivery methods, and therapeutic outcomes in MI. These articles were selected from the PubMed results as ordered by its internal relevance algorithm, which prioritizes recent publication date, citation frequency, and keyword relevance.

Based on findings in the selected articles, additional PubMed searches were conducted to explore future directions. The following queries were used: (“CRISPR” OR “gene editing”) AND (“stem cells” OR “iPSCs”) AND (“cardiac repair” OR “myocardial infarction”), as well as (“3D bioprinting” OR “bioprinted cardiac patches”) AND (“stem cells” OR “myocardial infarction”). Relevant articles were screened and selected for their potential contributions to advancing myocardial infarction treatment strategies.

## 3. Types of Stem Cells Currently Researched for MI Recovery

The types of stem cells that have been widely researched for MI recovery therapy are varied. Many studies focus on investigation of obvious stem cell candidates for myocardium regeneration, such as Cardiac Stem Cells (CSCs) [9,10,11], as well as the widely studied Mesenchymal Stem Cells (MSCs) [12,13,14,15,16] and Bone Marrow-derived Mononuclear Cells (BMMCs) [14,17,18,19,20,21]. Fewer studies focus on other stem cell types, ranging from Induced Pluripotent Stem Cells (iPSCs) [22] and Adipose-derived Stem Cells (ASCs) [23], to Cortical Bone-derived Stem Cells (CBSCs) [24], Hematopoietic Stem Cells (HSCs) [13] and other progenitor cells.

Among these stem cell types, distinction is also made between the genetic origins of the cells, with the two clinical and preclinical classifications generally being autologous or allogeneic origin. Autologous stem cells cultured from patient/model-derived tissue generally have decreased risk of immune rejection, but face issues related to the time needed to expand ex vivo (usually resulting in an infusion time post-acute phase) and the decreased potency of stem cells with age or disease. Allogeneic stem cells, on the other hand, have the ability to be cultured and ready “off-the-shelf” for infusion in the acute or subacute phase of MI recovery, and can be screened for the highest viability and/or potency, but run into aforementioned risks with Graft-vs-Host Disease (GVHD) and immune system inflammation [15,16].

The exact mechanisms through which stem cells affect the repair of cardiac function and structure are unknown, though two main mechanistic routes are thought to be at play: direct cardiomyocyte differentiation and paracrine signaling [9]. While inherent challenges to stem cell engraftment in host heart tissue arise due to the non-static cardiac environment and the “washing away” of transplanted cells by blood flow, a few key stem cell lineages show enhanced ability to successfully traffic and differentiate to healthy cardiomyocytes [11,22,25]. As for paracrine effects, the most significant paracrine signaling pathways associated with stem cell-mediated cardiac repair involve direction of angiogenesis [11,13,22], as well as a reduction in fibrotic scarring.

The following sections contain a description of the most relevant types of stem/progenitor cells currently researched for cardiac repair.

Figure 1 [26] briefly outlines the common origins of the different stem cell types found in the literature for cardiac repair.

### 3.1. Cardiac Stem Cells (CSCs)

Cardiac stem cells (CSCs) are a population of multipotent precursor cells (approximately 2% of total cardiac tissue) [27] within the heart that can differentiate into cardiomyocytes, endothelial cells, or smooth muscle cells. CSCs direct cardiogenesis during embryonic development, as well as minorly contribute to cardiac repair. Due to the antiproliferative nature of the cardiac environment and their low numbers, CSCs in adult tissue are limited in their natural reparative ability.

Though not found in the heart in sufficient quantities to direct considerable repair, when isolated from heart tissue, CSC therapy involving ex vivo expansion can raise their numbers to an amount suitable for transplantation. Notably, due to the process by which heart tissue is isolated and prepared, autologous origins of these CSCs are not feasible, instead requiring the remains of heart transplant atria in humans [11] or minced whole donor hearts in in vivo models [28] for allogeneic transplantation. While this raises the potential for GVHD complications, studies investigating CSCs largely report no significant risk of Major Adverse Cardiac Events (MACE) post-transplantation [10].

Hong and colleagues’ [9] exploration of c-kit+ cardiac stem cells found that after only 5 min-post intracoronary injection in mice, less than 40% of the initially injected CSCs were present in the cardiac tissue, and 24 h post-injection, less than 6% of the initial dose was present. Despite these findings, MI-induced mice treated with the CSCs had significantly lower Left Ventricular Diastolic Pressure, as well as a higher Left Ventricular Ejection Fraction (LVEF), indicating recovery towards pre-infarction heart function. This indicates that despite a lack of longevity in transplanted tissue, the paracrine effects of the CSCs significantly contribute to its ability to mediate cardiac repair.

Further exploration by Avolio and colleagues supported this mechanism, and found upregulated levels of VEGF, HGF, and FGF in CSC-transplanted murine models post-MI [11]. Interestingly, the study also indicated that a combinatorial treatment alongside Saphenous Vein-Derived Pericytes (SVPs) bolstered the treatments’ overall efficacy, indicating possible synergistic modalities between CSCs and other progenitor/stem cells that future research could investigate.

### 3.2. Mesenchymal Stem Cells (MSCs)

By far one of the most heavily researched classifications of stem cells, Mesenchymal Stem Cells (MSCs) are multipotent stromal stem cells that can be isolated from a variety of tissues, including bone marrow, adipose tissue, and umbilical cord tissue [29]. MSCs also have the ability to differentiate into a wide range of lineages, ranging from chondrocytes, cardiomyocytes, and even neurons [29,30].

Despite (or possibly due to) its differentiation abilities, MSCs’ niche in myocardial repair likely lies more within its paracrine signaling abilities and/or ability to recruit endogenous stem cells for repair rather than direct MSC-to-cardiomyocyte differentiation [12]. Studies have implicated it in repair mechanisms independent of CD34^+^ and CD45^+^ markers, suggesting a more paracrine mode of action associated via decreased collagen deposition and angiogenesis [13].

Given their ubiquity throughout the body and persistence ex vivo, both autologous and allogeneic MSCs have been investigated for MI recovery and have similar levels of efficacy. Whether allogeneic or autologous, studies investigating their role in post-MI recovery indicated good safety outcomes [16], though it should be noted that MSCs do have an established tumorigenic risk [31].

### 3.3. Bone Marrow Mononuclear Cells (BMMCs)

Not technically a homogenous stem cell population, Bone Marrow Mononuclear Cells (BMMCs) are a population of progenitor/multipotent stem cells found in bone marrow. Its population broadly includes Hematopoietic Stem Cells (HSCs), Endothelial Progenitor Cells (EPCs), and Mesenchymal Stem Cells (MSCs) [32], among others. Despite being an amalgam of progenitor cells and not a unique stem cell type, it is one of the most studied types of cell treatment/therapy, and thus warrants its own discussion.

Because of its varied cellular makeup, BMMCs exhibit a broad differentiation potential. They exhibit the ability to differentiate into all the lineages available to their constituent cells. Beyond this, they have some of the quickest turnaround times due to a lack of a need for significant ex vivo culturing, offering the rare possibility for autologous acute post-MI treatment [32].

Interestingly, and possibly due to its makeup being “dilute” with non-relevant progenitor cells, most studies which focused on functional markers of post-MI BMMC therapy found insignificant, if any, improvements in key markers such as LVEF, LVEDV, or LVESV, though it is important to mention a significant absence of MACE and other risk factors [17,19,20]. Despite this, BMMCs remain a promising candidate for stem cell therapy for MI recovery, as other studies indicate a synergistic effect of BMMCs in conjunction with homogenous stem cell populations such as MSCs [14] or when paired with Coronary Artery Bypass Grafts (CABGs) [21].

### 3.4. Induced Pluripotent Stem Cells (iPSCs)

Induced Pluripotent Stem Cells (iPSCs) are adult somatic cells that have been engineered back to an embryonic pluripotent state via forced gene expression and factor expression. They can be isolated from virtually any differentiated adult somatic tissue, though the most common sources are skin-derived fibroblasts or peripheral blood [33].

Given their pluripotent nature, they are able to differentiate into all cellular lineages and hold the highest differentiative potential of any stem cell alongside Embryonic Stem Cells (ESCs). Their ability to minimize donor-specific variability also makes them promising for both autologous and allogeneic therapeutic applications [34].

However, the very pluripotency that makes iPSCs advantageous also presents challenges, notably the risk of teratoma formation following transplantation [34]. As such, utilization of iPSCs as a starting population for differentiation into CSCs or MSCs before transplantation has been proposed to maximize the benefits while minimizing risk.

Within the context of cardiac repair post-MI, iPSCs have shown unique advantages in preclinical settings. Yang and colleagues’ transcriptome analysis of iPSC culturing in both a 2D and 3D environment clearly demonstrated successful differentiation into cardiomyocytes, as well as expression of paracrine signals to direct angiogenesis and recovery [22]. Should tumorigenic risk be mitigated, these preliminary studies indicate that iPSCs might be a strong candidate for a homogenous stem cell population therapeutic approach with the broadest potential effects.

### 3.5. Hematopoietic Stem Cells (HSCs)

Hematopoietic Stem Cells (HSCs) are a type of stem cell derived primarily from bone marrow and peripheral blood, which display multipotency in all blood lineages [35].

Hematopoietic stem cells (HSCs) can be safely obtained from both allogeneic and autologous sources, with annual therapeutic transplants being roughly evenly divided between the two origins (47% and 53%, respectively) [35].

Due to their hematopoietic lineage commitment, while HSCs do not exhibit direct cardiomyocyte differentiation capability, preclinical data indicates that their overarching paracrine effects on the heart are considerable. Shalaby and colleagues’ study in murine MI models found significant histopathological recovery associated with HSC treatment, as indicated by decreased collagen deposition and improved cardiac tissue architecture, as well as a significant upregulation in key paracrine signaling markers. Furthermore, infarct size reduction was even greater with HSC therapy compared to MSC treatment [13].

### 3.6. Adipose-Derived Stem Cells (ASCs)

Adipose-derived stem cells (ASCs) are a subset of mesenchymal stem cells (MSCs) obtained from adipose tissue. Like other MSCs, they exhibit multipotency, and can differentiate into various cell types, including osteogenic, chondrogenic, and adipogenic lineages. ASCs share many functional similarities with bone marrow-derived MSCs (BM-MSCs), but offer the advantage of being more readily accessible and available in higher quantities from fat tissue. While their regenerative potential and paracrine signaling effects largely overlap with other MSC populations, some studies suggest ASCs may have distinct immunomodulatory properties, making them a promising candidate for cell-based therapies [36].

Additionally, early studies on ASCs for MI recovery therapy have shown promising results in animal models, suggesting that their therapeutic potential is at least comparable to MSCs derived from other tissues [23].

### 3.7. Other Stem Cell Types

Other translationally relevant stem cell types within the current body of knowledge are Multilineage-Differentiation Stress-Enduring Cells (Muse Cells) and Cortical Bone-Derived Stem Cells (CBSCs).

Additionally, while Embryonic Stem Cells (ESCs) exist as a therapeutic avenue and have shown some preclinical promise, due to the ethical concerns regarding obtaining the cells, have not been investigated thoroughly or clinically [37].

Muse Cells are endogenous pluripotent stem cells present in various tissues such as bone marrow. As opposed to iPSCs, which must be artificially engineered to be pluripotent, Muse cells exist endogenously and exhibit similar pluripotency. Key preclinical studies implicated their ability for increased engraftment ability, immunomodulation, and angiogenesis at the level of or above that of traditional MSC or BMMC treatments [25]. However, clinical relevance is still unknown, and autologous therapeutic approaches are limited due to a long isolation and culturing timeline.

CBSCs are a novel somatic stem cell that in some preliminary in vivo and in vitro studies, has been implicated in cardiac repair and remodeling. They are derived from the cortical bone of donors prior to culturing, expansion, and transplantation [38]. Though exhibiting similar cell morphology to MSCs, CBSCs are unique in their WFA-binding glycan expression profile, evading glycan alterations that reduce differentiation abilities [24]. Moreover, the initial research indicates an immunomodulatory effect that reduces inflammation via promotion of regulatory T-Cells (Tregs).

A summary of the relevant stem cell types and important characteristics are listed in Table 1.

## 4. Role of Stem Cells in Cardiac Regeneration Post-Myocardial Infarction

As discussed previously, MI results in the death of cardiac myocytes and scar formation in the areas where myocyte death occurred [24]. With the emergence of stem cell therapy as a promising treatment option for MI, it is important to review the various contributions to cardiac regeneration employed by stem cells such as angiogenesis, paracrine signaling, and extracellular vesicles (EVs). These three mechanisms are explored in more detail below.

### 4.1. Restoration and Angiogenesis

Stem cell therapy holds great potential for cardiac regeneration, and restoration of cardiac tissue through angiogenesis is a key mechanism through which stem cells promote their regenerative effects. Angiogenesis is the proliferation of blood vessels from pre-existing vessels, and is crucial to increasing blood flow, especially to ischemic tissue which needs more blood flow to survive. Neoangiogenesis, a specific type of angiogenesis, can improve the effectiveness of coronary circulation, and can be triggered by myocardial infarction (MI), thereby reducing cardiomyocyte death and limiting the formation of non-contractile scar tissue [39]. MSC differentiation to cells such as myocytes, endothelial cells, and more could be a key contributor to angiogenesis. The induction of angiogenesis has been theorized to lead to more effective left ventricular function, a decrease in heart tissue damage, and the formation of new capillaries to help improve oxygen supply to cardiomyocytes (CMs) [16]. The main role of angiogenesis in cardiac regeneration post-MI could be the increased oxygen output to affected CMs. This could allow for the reduction in infarct size in the host heart, and thereby promote better recovery of heart function. Also, studies show that an increase in angiogenesis is directly related to the reduction in myocardial damage or infarct size. Multiple types of stem cells have been shown to increase angiogenesis and the effect of angiogenesis on cardiac regeneration. Some of these stem cell types include BMMSCs, MSCs, HSCs, EPCs, iPSCs, CSCs, and more. The level of angiogenesis from each stem cell type could vary, which could mean that some stem cells are more suited for promoting restoration and angiogenesis. The way stem cells can exert angiogenesis is through secreting different proangiogenic factors. There are multiple proangiogenic factors, some of which include vascular endothelial growth factor (VEGF), basic fibroblast growth factor (FGF), hepatocyte growth factor (HGF), IGF-1, tissue growth factor-β (TGF-β), and angiopoietin-1 [40]. Overall, angiogenesis is a key biological process important for heart restoration post-MI, and multiple factors contribute to angiogenesis promotion that could be highlighted in future studies, such as the role of growth factors, proangiogenic factors, and different signaling pathways.

### 4.2. Extracellular and Paracrine Signaling

Once stem cells are implanted into the body, they exert their regenerative effects, mainly through mechanisms like paracrine signaling rather than direct differentiation into new cardiac cells. Paracrine factors like chemokines, growth factors, and cytokines play a big part in heart regeneration through mechanisms, such as regulating inflammatory responses, increasing the level of angiogenesis, attracting naturally present stem cells to the damaged area, and promoting cardiac cells to resume the cell cycle [41]. The paracrine hypothesis is a term used to describe paracrine effects compared to direct stem cell differentiation for stem cell therapy and their overall effectiveness on cardiac regeneration. The chemical signals that are released by implanted stem cells stimulate a multitude of regenerative mechanisms. This includes angiogenesis stimulation, lowering the level of tissue hypertrophy, and apoptosis of nearby cardiac cells, extracellular matrix (ECM) remodeling, and the stimulation of CSCs [40,41]. Paracrine factors like VEGF, IL-6, MCP-1, PGF, and the FGF family are key for angiogenesis stimulation. FGF21 and FGF23 are some of the specific genes from the FGF family that interact with receptors on CMs, playing an important role in prevention against hypertrophic remodeling and regulating cell proliferation, which are some of the mechanisms that contribute to cardiac damage post-MI. WNT signaling, a paracrine pathway that has shown potential in cardiac regeneration, plays an important role in the growth of the left ventricle and CM differentiation [22,42]. TGF-β1 is another cytokine released from stem cells with a theorized strong paracrine effect once implanted. The cytokine stimulates the activation of fibroblasts, allowing them to potentially contribute to cardiac regeneration post-MI. Other paracrine factors contribute to the regeneration process by recruiting nearby cells in the host heart to the injury site, helping facilitate the repair process and potentially increasing the repair speed and efficiency [24]. Overall, paracrine signaling from stem cell therapy shows immense potential in the future to be a main contributor to cardiac regeneration.

### 4.3. Extracellular Vesicles (EVs) and Their Role in Cardiac Regeneration

Extracellular vesicles (EVs) are important in mediating stem-cell based effects on cardiac regeneration and repair. EVs are cell-derived membrane vesicles that have differing diameters. They transport a variety of biomolecules such as membrane proteins, cytosolic proteins, metabolites, and nucleic acids. Some of the important nucleic acids present in EVs are long RNAs that are non-coding, mRNAs, and microRNAs [43]. Based on these components present in EVs, the ones that are derived from stem cells likely have the function of gene expression regulation. There are three classifications of EVs, with the classification based on origin: apoptotic bodies that come from cells on the verge of death, microvesicles (MVs) that are formed by budding vesicles off the plasma membrane, and exosomes that are released from the endosomal pathway and multivesicular body fusion [44]. Exosomes could be the type of EV that shows the most potential in cardiac regeneration as they can come from a variety of stem cells and have a low chance of developing into harmful tumors. They also promote CM proliferation, angiogenesis, anti-inflammatory responses, lowering the infarct size, increasing stem cell activity of the native heart, reducing the level of apoptosis for local CMs, and metabolic regulation through gene expression modulation. Studies show positive effects from exosomes on cardiac function, especially those derived from CPCs, hiPSCs, and MSCs. The positive exosome effect is different for each of the stem cells mentioned. Ischemic preconditioning is also an important process that can increase the stimulation of exosome release and increase their levels in the myocardium, with the process in MSCs increasing the level of miRNAs (miR-21, miR-22, miR-199a-3p) that play a role in fibrosis and apoptosis reduction. Exosomes also show potential in CM survival and decreased inflammation after blood flow recovery. Exosomes have sustained reliability and low immune activation to the transplantation, but also require multiple injections and specific cellular recognition to exert their effects on the heart. The exosomes injected may contain components other than the desired ones that may lead to unwanted effects other than cardiac regeneration [45]. Small extracellular vesicles (sEV) are also shown to play a role in cardiac repair and regeneration post-MI, with sEV derived from human embryonic stem cells (ESV) showing some of the best pro-angiogenic and scar-reducing properties. This is due to the fact that sEV from ESV are rich in fibroblast growth factor-2 (FGF-2), a molecule that promotes mitotic activity and angiogenic processes [43]. These mechanisms that EVs have to contribute to cardiac regeneration further highlight the potential they have in MI treatment.

## 5. Methods and Timing of Delivery

Optimal delivery method for stem cell administration heavily depends on several factors, such as whether the patient will undergo open thoracic surgery or cardiac catheterization, the presence of diffuse significant stenosis in coronary vessels, and whether the patient has recently experienced a myocardial infarction, which may produce homing signals that direct stem cells to specific regions of the heart. Determining the target area for stem cell delivery is equally critical. For example, the cells can either be localized to a single defined coronary artery or venous territory of the heart that sustained damage after an acute infarct, or they can be more broadly dispersed to support more than one specific area.

Of note, myocardial infarctions are broken into three major time periods. Within 7 days of the MI is referred to as the acute phase, which has the highest risk of complications and death. Between the 7th and 14th day post-MI is the subacute phase, which is characterized by granulation and clearing of damaged tissue by macrophages. After 14 days, the time period is referred to as the post-acute chronic phase, which is characterized by cardiac scarring and remodeling. With the complex changes occurring throughout the myocardium in each phase, the timing of the delivery is also an important factor in the administration of cardiac stem cells in order to yield the best results. The 2006 REPAIR-AMI Trial found that bone marrow mononuclear cells administered within 4 days of myocardial infarction were not superior to placebo with regard to improving left ventricular function, which is thought to be due to toxic products of reperfusion and associated tissue edema, which prevent introduced cells from surviving [46]. Furthermore, a 2015 study found that, compared to administration 1–3 days, 4–7 days, 8–14 days, and 15–30 days after MI, there was the largest increase in left ventricular function and fewest adverse cardiac events when stem cells were administered between 4 and 7 days after MI [47]. A follow-up study in 2017 agreed with these findings, and concluded that cell administration on day 3 post-MI is superior to administration within 24 h or over 7 days post-MI in increasing left ventricular ejection fraction and decreasing end-systolic and end-diastolic measurements of the left ventricle [48]. Studies that did not find any difference in outcomes between days of administration on left ventricular function, such as the TIME Trial, SWISS-AMI Trial, and LateTIME Trials also found no increase in left ventricular function regardless of administration day [49]. Thus, based on the available studies demonstrating an increase in left ventricular function, the optimal window of administration of stem cells appears to be within the acute phase around day 3–7 after MI.

The specific method chosen by the treatment team should be evaluated thoroughly since the cell delivery method affects cell retention, survival, integration, and functionality. Several methods have been investigated, and each presents unique advantages and disadvantages. These methods include the following.

### 5.1. Direct Intramyocardial Injection

Direct intramyocardial injection involves targeting specific infarcted areas of the myocardium, which are identified by nuclear imaging and/or cardiac echocardiography [50,51]. Stem cells are injected directly into the infarcted region during thoracotomy in open-heart surgeries, such as coronary artery bypass grafting (CABG) [52] or as independent procedures, including lateral mini thoracotomies [53].

Under direct intramyocardial injection, there is a high cell retention and engraftment rate within the infarcted area, as it does not require mobilization and homing of the stem cells to the area of interest. It has also been shown to significantly improve left ventricular ejection fraction and improved cardiac function [14,52,54]. However, as with all open procedures, it is highly invasive, and requires a surgical approach with significant procedural risks and potential for complications, including arrhythmias, myocardial perforation, and systemic embolization [51,55,56]. In addition, it is limited to specific areas of the myocardium, and can potentially leave other affected regions untreated. The recovery period is also quite long, and is not suitable for repeated use in the same individual. Implanted cells can also be lost from the site of injection [57].

### 5.2. Intravenous Infusion

Intravenous infusion is primarily used after following acute myocardial infarction, since the success is heavily dependent on signaling molecules released by the dying myocardium [58]. This method is attractive, since it is minimally invasive and easier to administer compared to other methods. It is also particularly useful in cases involving multiple infarct areas, as the stem cells will travel to any infarcted areas of the heart. However, cells are frequently trapped in the lungs [59], leading to low delivery efficiency, low retention in the myocardium [60], and reduced therapeutic efficacy. There is also less improvement in left ventricular ejection fraction compared to direct myocardial injection.

### 5.3. Intracoronary Infusion

Intracoronary infusion utilizes standard balloon coronary catheterization catheters to deliver stem cells through specific coronary arteries [61]. The catheter is used to access the coronary artery of interest via a transfemoral approach, and the stem cells are injected under either stop-flow conditions, with balloon occlusion performed to minimize the cells from entering the systemic circulation [62] or under continuous-flow conditions, with ongoing coronary blood flow [63]. Studies have shown improvement in left ventricular function, reduction in adverse ventricular remodeling, reduction in infarct size, and improvement in clinical outcomes, as well as quality of life [64,65,66].

Additionally, intracoronary infusion facilitates direct administration of stem cells to the affected area and increased cell retention in the myocardium compared to intravenous infusion. Infusing the stem cells via intracoronary infusion is also convenient, since it can be completed during routine cardiac catheterization procedures, which are already a vital procedure for diagnosing coronary occlusions and MIs [66]. Despite these benefits, there is still potential for coronary artery damage, microvascular obstruction, and embolization [67,68], which limits the size and dose of stem cells delivered using this approach [69]. Furthermore, it is difficult to deliver stem cells to areas that are not perfused well or have more delicate blood supply through smaller vessels. Cell survival can also be affected by hypoxia and nutrient deprived conditions in the affected myocardium. Stem cells may also undergo myocardial trapping and redistribution in the pulmonary vasculature, despite bypassing circulation through the lungs before reaching the heart [70].

### 5.4. Intramyocardial Administration Using Catheters

Intramyocardial administration using catheter guidance mitigates issues posed by occluded coronary arteries, which makes it an excellent option for use in patients with congestive heart failure (CHF) secondary to ischemic heart disease. There are two catheter guided intramyocardial administration approaches: transendocardial and transcoronary injections.

#### 5.4.1. Intramyocardial Trans Endocardial Injection

Kornowski and colleagues were the first to employ this method in a swine model, and demonstrated marked improvement in cardiac function [71]. The method described by Kornowski involved using electromechanical mapping to direct delivery of stem cells at the border zone between necrotic and healthy endocardium [68,71,72]. This technique has shown significant improvement in left ventricle ejection fraction and infarct size reduction in both clinical and preclinical studies [71,72]. It also allows precise delivery of cells to the infarcted area with few significant adverse complications, which gives it high efficacy and high rate of cell retention. However, specialized equipment and expertise is required, and while reduced, there are some procedural risks, including myocardial perforation, arrhythmias, and inflammatory responses [73,74].

#### 5.4.2. Intramyocardial Trans Coronary Venous Injection

Thompson and colleagues first described this method using intravascular ultrasound imaging in a swine model [75]. In this approach, the coronary veins are directly accessed and the stem cells are injected.

Intramyocardial transcoronary venous injection allows homogeneous distribution across the myocardium and high retention of stem cells [76]. It is also a minimally invasive procedure, and can be performed percutaneously [75], which allows for targeted delivery to specific areas of the myocardium [75,77] without the risks associated with open procedures. Unfortunately, there is significant anatomic variability in the coronary venous system, and there may be difficulty in stem cell delivery to the correct coronary territory. It also carries the risks of arrhythmogenicity, inflammatory response, mechanical loss of the cells, and pulmonary redistribution [73,74].

### 5.5. Retrograde Coronary Venous Delivery System

Retrograde coronary venous delivery involves advancing a catheter through the femoral vein into the right atrium, followed by cannulation of the coronary sinus. The cardiac veins are accessed, and a balloon is inflated to prevent access to systemic circulation. Once the treatment area is isolated, stem cells are injected [78]. This method is beneficial in cases of coronary artery obstructions, and in patients who are not good candidates for coronary artery bypass graft [78,79]. There is also lower risk of coronary embolism compared to intracoronary injection. Despite the pros, it remains difficult to navigate the coronary venous system due to its tortuosity [80], and there is risk of damage to the coronary sinus, which can be complex and difficult to repair [81].

## 6. Summary of Clinical and Preclinical Trials

Among the various stem cell therapies for myocardial infarction (MI) mesenchymal stem cells (MSCs) and autologous cardiosphere-derived cells (CDCs) showed the most promise. Intracoronary (IC) infusion of MSCs post-percutaneous coronary intervention increased LVEF by 8.8% at four months after injection, while the control increased LVEF by 4.8% (*p* = 0.031) [82]. A booster dose of Wharton’s Jelly Mesenchymal Stromal Cells (WJ-MSCs) improved LVEF function by 7.45%, while a single dose increased LVEF by 4.54% [13].

Additionally, IC CDC infusion significantly reduced scar size (−11.1%) and increased viable myocardium (22.6%) in the one year follow-up [83]. BM-MNCs delivered through graft vessels during Coronary Artery Bypass Grafting (CABG) enhanced left atrium function compared to sole usage of CABG [83]. Overall, there were no significant safety concerns or adverse events associated with the stem cell injections.

An overview of the key clinical and preclinical trials are listed in Table 2.

## 7. Limitations in Cell Survival and Retention

Only a very small fraction (around 1% or less) of transplanted stem cells survive long-term after transplantation. Ischemia, inflammation, and mechanical washout by the myocardium are the largest contributing factors to the substantial cell loss [6,16]. Intracoronary or intravenous cell infusions have been known to suffer from rapid washout, which limits effective cell engraftment. Novel strategies such as direct intramyocardial injections, along with the usage of injectable hydrogels and epicardial patches have been developed to improve acute cell retention by creating a supportive extracellular matrix environment [37,41]. Injectable hydrogels have shown up to 8–14× increased retention, while epicardial patches have shown up to 47–59× increased retention, compared to saline controls [41]. Due to difficulties trying to achieve high survival rates for implanted cells, multiple studies have advocated for the usage of cell derived exosomes (sEVs) as an alternative treatment option. Such vesicles are able to carry reparative signals, and avoid issues associated with cell viability [43].

### 7.1. Tumorigenicity Risks

Therapies that particularly involve pluripotent stem cells, such as ESCs and iPSCs, pose inherent risks of uncontrolled cell proliferation and teratoma formation if undifferentiated cells are transplanted into unintended regions [91]. Even when properly transplanted towards a cardiac lineage, the presence of immature or incompletely differentiated cells may lead to arrhythmias or neoplastic lesions. Autologous BM-MSCs possess a lower risk of tumorigenicity due to the fact that they are lineage-restricted, and that they possess a lower likelihood of forming teratomas. However, extensive in vitro expansion may alter their phenotype, and potentially raise risks [88]. Usage of cell-derived exosomes are highlighted as a promising alternative to circumvent tumorigenic risks due to the fact that they are non-replicative, as they do not possess the appropriate cellular framework necessary for uncontrolled cell growth [43,92]. Advances in gene editing, such as techniques like CRISPR/Cas9 in hPSC-derived cardiomyocytes, can help reduce residual pluripotency and off-target mutations, hence lowering tumorigenic potential [93].

### 7.2. Fibrosis and Adverse Remodeling

Following myocardial infarction, the loss of cardiomyocytes is replaced with fibrotic scar tissue, which leads to adverse ventricular remodeling and impaired function [88]. BM-MSCs and other stem cell therapies rely on paracrine signaling, as they secrete cytokines and growth factors such as VEGF and adrenomedullin that potentially mitigate fibrosis, and promote angiogenesis [41,94]. In clinical settings, improvements in terms of reducing fibrotic scar or adverse remodeling have been modest at best [41]. Cardiac patches engineered from stem cells or biomaterials are currently being explored to provide structural support and reduce fibrosis by promoting increased cell-survival and organized tissue regeneration [83,88]. ESC-derived exosomes demonstrate potent anti-fibrotic effects in vitro, reducing markers such as collagen and α-SMA, and also in vivo, by promoting angiogenesis, hence counteracting adverse remodeling. Though improvements in surrogate endpoints, such as LVEF (ex. 48–57% at 12 months) have been noted in certain trials, many studies have not yet demonstrated clear, consistent, and sustained reductions in adverse remodeling, measured by LVEDV, LVESV, and infarct size changes, highlighting the need for further research into cell therapy protocols [16].

## 8. Future Stem Cell Technologies/Treatments

### 8.1. Gene Editing

The convergence of gene editing with stem cell technology has opened new avenues for cardiac regeneration. The following will be specific research directions and evaluation of the most viable approaches for stem cell therapy in cardiac applications.

Combining different stem cell types to create cardiac stem cell hybrids is an emerging strategy aimed at harnessing the complementary properties of each cell type to enhance myocardial repair. With Mesenchymal Stem Cells (MSCs) and Cardiomyocytes, hybrid constructs combining MSCs with cardiomyocytes have been developed to provide structural support and promote functional integration into the host tissue. With Endothelial Progenitor Cells (EPCs) and Cardiomyocytes, co-culturing EPCs with cardiomyocytes can enhance vascularization of the engineered tissue, improving oxygen and nutrient delivery post-transplantation [93].

### 8.2. Small Extracellular Vesicles

Extracellular vesicles, particularly sEVs, emerge as mediators of intercellular communication. sEVs derived from iPSCs carry bioactive molecules such as proteins, lipids, and RNAs that can modulate recipient cell behavior. iPSC-derived sEVs have demonstrated potential in promoting cardiac repair by enhancing angiogenesis, reducing apoptosis, and modulating immune responses, and upregulating pro-angiogenic factors, facilitating the formation of new blood vessels in ischemic myocardial tissue. In addition, the bioactive cargo within these vesicles can inhibit cardiomyocyte apoptosis, thereby preserving cardiac function post-injury [44]. Many studies suggest that the therapeutic effects of stem cells are largely mediated by their secreted factors rather than direct differentiation into functional cardiac cells. sEVs act as carriers of bioactive molecules such as microRNAs, proteins, and lipids, which can influence recipient cell behavior and trigger regenerative responses in damaged myocardium. One major limitation of direct iPSC transplantation is the risk of teratoma formation, as iPSCs have unlimited proliferative capacity. Since sEVs are acellular vesicles, they do not carry the risk of tumor formation or uncontrolled differentiation, making them safer for clinical application. Examining the use of sEVs from iPSCs to promote cardiac repair could be a viable path due to their relative stability and comparatively easier storage to live cells [44,89].

### 8.3. Bioinks and Bioprinting

The efficacy of stem cell therapy for cardiac diseases is significantly influenced by the delivery method. Poor retention, immune rejection, and limited integration remain key challenges that optimized delivery methods aim to address. The harsh post-infarction environment—marked by inflammation, hypoxia, and mechanical stress—leads to significant loss of transplanted cells. Intramyocardial injection, intracoronary infusion, and intravenous administration work in specialized settings depending on the situation employed, but are limited by lack of research and specialized equipment [95,96]. Delivery methods that improve cell engraftment and integration can significantly enhance the effectiveness of cardiac regeneration, which can be a viable direction for technology. The current market direction entails bioengineered scaffolds which support stem cell attachment, proliferation, and differentiation while providing a microenvironment that mimics native myocardium. Rapid progression and an emerging importance of 3D bioprinting in cardiac regenerative medicine is a good avenue to explore specific delivery methods and electrospun nanofibers as delivery vehicles, which provide a matrix that guides cell alignment and promotes electrical coupling between transplanted and native cardiomyocytes [89].

Three-dimensional bioprinting for targeted cell placement especially enables precise cell deposition to create structured cardiac patches that enhance integration. Several bioprinting techniques have been explored for cardiac tissue engineering. Extrusion-Based Bioprinting, for example, allows for the deposition of continuous filaments of bioink, enabling the construction of complex, multicellular structures. Cardiac repair requires thick, multicellular constructs capable of withstanding contractile stress and maintaining paracrine signaling over time. Extrusion-based bioprinting enables high cell densities, scaffold thickness, and spatial patterning of supporting materials (e.g., ECM analogs like dECM). This directly addresses the issue of low cell retention and survival in injected cell therapies (e.g., Gnecchi et al., 2006 [94]). Moreover, extrusion printing allows inclusion of layered gradients, mimicking myocardium’s anisotropic architecture—thus supporting functional conduction and contractility across the construct. This capability is indispensable in building patches that not only survive, but also synchronize with host tissue, a major milestone for myocardial infarction repair [97].

Inkjet Bioprinting utilizes droplets of bioink, and offers high resolution but is limited to low-viscosity materials. Inkjet methods contribute by enabling microscale delivery of signaling molecules and cells in precise geometries—crucial for angiogenesis and spatiotemporal paracrine control. In cardiac settings, this allows selective deposition of endothelial cells or angiogenic factors like VEGF in capillary-like patterns that guide vascular self-assembly post-implantation. While not ideal for bulk tissue, this technique supports the bioactive coating of engineered patches, ensuring faster perfusion, oxygenation, and immune modulation once implanted. Thus, it complements macro-scale constructs by enhancing integration, not just transplantation [98].

Laser-Assisted Bioprinting also provides precise cell placement and high resolution, which is suitable for creating intricate tissue architectures. These techniques are examples that facilitate the fabrication of cardiac patches with organized structures, promoting better integration and function post-implantation. Laser-assisted bioprinting (LAB) addresses creating microvascular structures at scales <100 µm—beyond the resolution of extrusion and inkjet. In cardiac tissues, where oxygen diffusion limits viability beyond 200 µm, LAB enables embedding of perfusion-ready capillary networks within printed patches. This is not cosmetic—it defines functional viability. Moreover, LAB allows printing of cardiomyocytes in precise orientations, aligning contractile units and mimicking native sarcomere architecture. In this way, LAB contributes to creating not just replacement tissue, but functionally mature, integrated myocardium [97].

In addition to this, bioinks are crucial for replicating the extracellular matrix (ECM) of cardiac tissue. Natural hydrogels like gelatin methacryloyl (GelMA), collagen, fibrin, and decellularized ECM (dECM) are commonly used due to their biocompatibility and ability to support cell viability and function. Synthetic polymers such as polyethylene glycol (PEG) derivatives can be tailored for specific mechanical properties but may require functionalization to enhance cell adhesion [98]. These techniques facilitate the fabrication of cardiac patches with organized structures, promoting better integration and function post-implantation.

### 8.4. Feasibility Studies

Case studies that employ these methods demonstrate reactivity and feasibility. Mydicar exemplifies the potential of combining gene therapy with cardiac treatment. It involves delivering the SERCA2a gene to cardiomyocytes using an adeno-associated viral vector, aiming to enhance calcium handling and improve cardiac contractility [99]. Clinical trials have demonstrated a 52% reduction in the risk of worsening heart failure among treated patients, highlighting the therapeutic potential of gene therapy in cardiac diseases. Human hepatocyte growth factor (HGF) possesses angiogenic and anti-apoptotic properties. Administering HGF plasmid DNA to ischemic cardiac tissue has shown promise in promoting angiogenesis and reducing adverse remodeling post-myocardial infarction. Clinical trials have indicated safety and potential efficacy, considering the viability of gene therapy approaches in cardiac regeneration [93,100].

Preclinical trials have also begun for the potential of 3D-bioprinted cardiac tissues in improving myocardial repair. For instance, combining mesenchymal stem cells (MSCs) with cardiac progenitor cells (CPCs) has shown enhanced myocardial repair in animal model [101]. However, clinical translation remains challenging. The REGENERATE-AMI trial, which investigated intracoronary delivery of autologous bone marrow-derived cells in acute myocardial infarction patients, did not show significant improvement in clinical outcomes at the five-year follow-up [102].

While clinical applications are still emerging, the advancements in 3D bioprinting technology and stem cell biology hold promise for the development of patient-specific cardiac patches. Ongoing research focuses on enhancing the maturation of engineered tissues, ensuring their mechanical and electrical integration with the host myocardium, and establishing robust vascular networks to support tissue viability post-implantation.

## 9. Conclusions

In this literature review, we identified that MSCs, BMMCs, CSCs, HSCs, iPSCs, and ASCs have been shown to increase paracrine signaling to straightforward acquisition. However, these therapies also face challenges such as low retention, tumorigenic risk, and immunologic complications. Among these, MSCs and CDCs demonstrate the most potential. Intramyocardial injection of MSCs is feasible and safe during five-year follow-up periods, resulting in improved LV function, but showing no significant differences in event-free survival compared to controls. Likewise, CDCs administered via intracoronary infusion have also been found safe and feasible, showing reductions in LVEDV, LVESV, and NT-proBNP at six months. However, no significant decrease in scar size was observed at either six or twelve months. Additionally, BM-MNCs delivered through graft vessels during CABG enhanced left atrial function compared to CABG alone. Nevertheless, long-term survival of transplanted stem cells remains limited, typically at around 1% or less.

To address limitations, emerging areas of research should focus on (1) combination of gene cell therapies, (2) delivery methods to deliver pro-regenerative signals without the risks associated with whole-cell transplantation, and (3) advanced gene-editing tools to mitigate tumorigenicity and improve cardiomyocyte differentiation efficiency. Further optimization in cell sourcing, delivery methods, and genetic manipulation is necessary to achieve consistent and durable clinical benefits. Continued collaborative efforts across regenerative medicine, bioengineering, and clinical research are essential to refine these therapies and advance them toward broader clinical implementation.

## Figures and Tables

**Figure 1 biomedicines-13-01209-f001:**
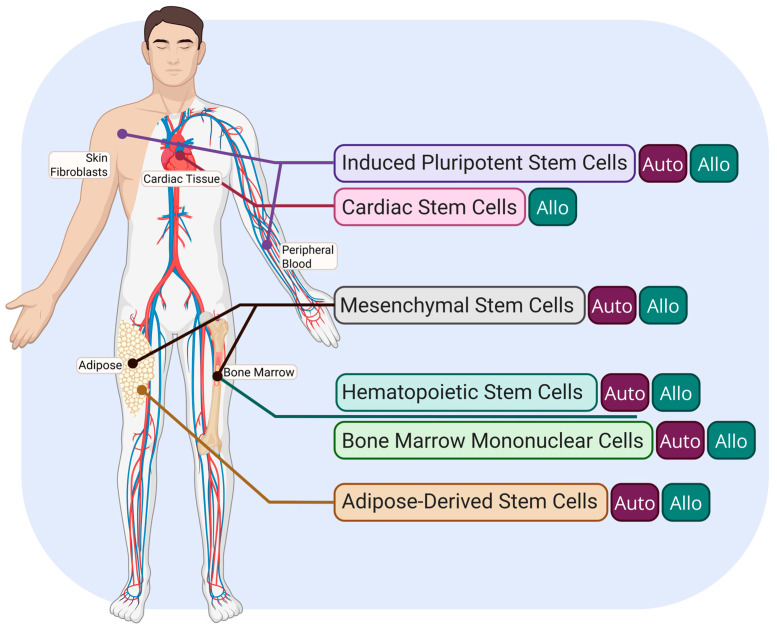
Overview of the most prominent stem cell population origins for post-MI therapies. Autologous origin is indicated by the “Auto” tag, and allogeneic origin is indicated by the “Allo” tag. Created in Biorender. Le. (2025) https://app.biorender.com/illustrations/67d20536719d6d5bbe39b236 (accessed on 30 March 2025).

**Table 1 biomedicines-13-01209-t001:** Summary of Clinical and Therapeutic Characteristics of Different Stem Cell Types.

Stem Cell Type	Common Cell Origin	Possible Lineages	Therapeutic Concerns	Reference(s)
Cardiac Stem Cells (CSCs)	Heart tissue	Cardiomyogenic, Endothelial, Smooth Muscle	Potential for GVHD (allogeneic use), low engraftment	[9,10,11,27,28]
Mesenchymal Stem Cells (MSCs)	Bone marrow, adipose tissue, umbilical cord	Osteogenic, Chondrogenic, Adipogenic, Myogenic, Tenogenic, Neurogenic	Tumorigenic risk	[12,13,16,28,30,31]
Bone Marrow Mononuclear Cells (BMMCs)	Bone marrow	Hematopoietic, Endothelial, Mesenchymal	Limited efficacy alone	[14,17,19,20,21,32]
Induced Pluripotent Stem Cells (iPSCs)	Skin-derived fibroblasts, peripheral blood	Ectodermal, Mesodermal, Endodermal	Teratoma formation risk	[22,33,34]
Hematopoietic Stem Cells (HSCs)	Bone marrow	Myeloid, Lymphoid	Low engraftment	[13,35]
Adipose-Derived Stem Cells (ASCs)	Adipose tissue	Osteogenic, Chondrogenic, Adipogenic, Myogenic, Neurogenic, Angiogenic.	Tumorigenic risk	[23,36]
Muse Cells	Various tissues (bone marrow, etc.)	Mesodermal, Endodermal, Ectodermal	Limited clinical relevance, long isolation time	[25]
Cortical Bone-Derived Stem Cells (CBSCs)	Cortical bone	Osteogenic, Chondrogenic, Adipogenic, Angiogenics	Limited established research	[24,38]
Embryonic Stem Cells (ESCs)	Embryos	Ectodermal, Mesodermal, Endodermal	Ethical concerns	[37]

**Table 2 biomedicines-13-01209-t002:** Stem Cell Therapies for Cardiac Repair: Study Designs, Delivery Methods, and Outcomes.

Type of Stem Cell	Study Type	Delivery Route/Method	Dosage	Population Size	Results	Reference
Wharton’s Jelly Mesenchymal Stromal Cell (WJ-MSC)	Clinical, Single-blind, Randomized, Multicenter Trial	Intracoronary injection (once vs. twice)	Single dose 10^7^ WJ-MSCs, Single (10^7^) + Booster dose (10^7^) = 20^7^ cells	N = 65	Baseline LVEF: 40% in all groups. Single MSC increased LVEF by 4.54 ± 2% (one dose), Booster dose increased by 7.45 ± 2% (*p* < 0.001). Echocardiography: 6.71 ± 2.4% (one dose), 10.71 ± 2.5% (booster).	[84]
Bone Marrow Mononuclear Cell (BM-MNC)	Clinical, Phase 3 Randomized Trial (3-Year Follow-Up)	Intracoronary infusion	Single dose of 10^7^ cells	N = 390	Trial incomplete; expected outcome: evaluate prevention of HF post-AMI.	[85]
Autologous Cardiosphere-Derived Cells (CDCs)	Clinical, Double-Blind, Placebo-Controlled Trial	Intracoronary (IC) injection	2.5 × 10^7^ cells	N = 134	IC infusion of CDCs (CAP-1002) was safe and feasible. No significant reduction in scar size at 6 or 12 months. LVEDV and LVESV reduced at 6 months, NT-proBNP decreased.	[86]
Bone Marrow-Derived Cells (BMCs) Autologous	Clinical, Randomized Controlled Trial (85 Patients, AMI)	Intracoronary infusion into infarct-related artery	19.8 × 10^7^ cells	N = 85	MACE incidence: BMCs (26.1%), Placebo (18%). No mortality, no significant difference in MI or revascularization between groups.	[83]
Bone Marrow Mesenchymal Stem Cells (BM-MSCs)	Clinical, Randomized, Single-Blind, Multicenter, Controlled Trial	Intracoronary infusion 15 days after PCI	3.31 × 10^6^ cells	N = 43	IC BM-MSCs did not improve LV function or myocardial viability at 12-month follow-up. Safe, no toxic events.	[87]
Bone Marrow Mesenchymal Stem Cells (BM-MSCs)	Clinical, Randomized, Open-Label, Multicenter Trial	Intracoronary administration into infarct-related artery (IRA) at one month	7.2 × 10^7^ cells	N = 69	LVEF by SPECT at 6 months: BM-MSC group (5.9% ± 8.5%) vs. Control (1.6% ± 7.0%, *p* = 0.037). No significant LVEDV or LVESV differences.	[88]
Bone Marrow Mesenchymal Stem Cells (BM-MSCs)	Clinical, Randomized, Single-Blind Trial	Intracoronary infusion 1 month post-PCI	7.2 × 10^7^ cells	N = 30	LVEF increased by 8.8 ± 2.9% (BM-MSC) vs. 4.8 ± 1.9% (control) at 4 months (*p* = 0.031). No increased adverse events. Echocardiography showed sustained improvement at 12 months.	[42]
Allogeneic Human Cardiac Stem Cells (AlloCSC-01)	Clinical, Multicenter, Randomized, Double-Blind, Placebo-Controlled Trial	Intracoronary infusion at days 5–7 post-STEMI	3.5 × 10^7^ cells	N = 49	No deaths or MACE at 12 months. Infarct size reduction: −2.3% (95% CI, −6.5% to 1.9%). No differences in ventricular remodeling. Low immunogenicity.	[10]
Bone Marrow-Derived Progenitor Cells (BMCs)	Clinical, Randomized, Multicenter, Placebo-Controlled Trial	Intracoronary infusion post-STEMI	2.0 × 10^7^–2.5 × 10^7^ cells	N = 54	At 12 months, BMC treatment effect on EF was 2.8% (*p* = 0.26). In patients with EF ≤ 48.9%, BMCs improved EF (+6.6%, *p* = 0.01), reduced EDV increase (*p* = 0.02), and prevented ESV increase (*p* = 0.01).	[82]
Mesenchymal Stem Cells (MSCs) vs. CD34^+^ Cells	Preclinical, Rat Model	Intravenous injection	2 × 10^6^ cells	N = 48	CD34^+^ cells showed superior efficacy over MSCs in infarct size reduction and angiogenesis markers (VEGF, VEGFR-2, Ang-1, Tie-2 upregulation). Both cell types improved myocardial tissue structure, but CD34^+^ had significantly better outcomes.	[13]
Mesenchymal Stem Cells (MSCs) vs. Bone Marrow Mononuclear Cells (BMMNC)	Preclinical, Porcine Model	Trans endocardial injection	10^7^ autologous MSCs or BMMNCs	N = 15	MSCs improved LVEF significantly over BMMNCs at 4 weeks post-injection. BMMNCs showed no significant LVEF improvement.	[14]
Mesenchymal Stem Cells (MSCs)	Preclinical, mouse model	Intracoronary infusion,	1.0 × 10^5^ cells	N = 30	Cardiac Stem Cell Hybrids Enhance Myocardial Repair. CCs improved left ventricular anterior wall thickness and increased capillary density.	[89]
Autologous Cardiosphere-Derived Cells (CDCs)	Clinical, Randomized, Controlled Trial	Intracoronary infusion	12.5 to 25 × 10^6^ cells	N = 17	CDC therapy led to reduced scar size, increased viable myocardium, and improved regional function at 1-year follow-up. No significant safety concerns observed.	[90]
Bone Marrow Mononuclear Cells (BMMNC)	Clinical, Randomized, Controlled Trial	Delivery through graft vessel during CABG	13.28 × 10^7^ cells	N = 42	CABG + BM MNC improved LA function more than CABG alone. LAGS, LVEF, and LAV significantly improved postoperatively. Two-dimensional strain imaging was a sensitive tool for LA function evaluation.	[21]
Mesenchymal Stem Cells (MSCs) (Autologous Bone Marrow-Derived Ex Vivo Expanded Mesenchymal Stem Cells)	Clinical, Randomized, Controlled Trial	Intramyocardial Injection	31 × 10^6^ cells	N = 54	Intramyocardial MSC injection in AMI patients was feasible and safe for up to 5-year follow-up. LV function improved, no significant differences compared to controls in event-free survival.	[16]

## Data Availability

No new data were created or analyzed in this study. Data sharing is not applicable to this article.

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
