# Peer review of "Stem Cell Therapy for Myocardial Infarction Recovery: Advances, Challenges, and Future Directions"

_biomedicines, 2025, doi:10.3390/biomedicines13051209_

Round 1

Reviewer 1 Report

Comments and Suggestions for Authors

The manuscript that I have been the pleasure to evaluate deals with the innovative stem cell therapy in patients with myocardial infarction.

The review background is well presented, with an exhaustive explanation of the different stem cells' types and delivery methods. Issues regarding the potential neoplastic cell differentiation are also reported and explained.   

No major issues have been detected about the review sections or contents. Minor issues and considerations/comments have been listed below:

  • In the Literature Review Methodology, the authors report that "the first 100 articles were screened and..". Which criteria has been chosen to make this selection? 100 recenter studies? 100 studies with the widest population? Something different? Please clarify. 
  • Direct stem cell injection after a percutaneous intervention for an acute myocardial infarction may represent the brightest therapeutic improvement for clinicians. However, as per revascularization timing, is there a therapeutic window for stem cell infusion? Are data currently available?
  • The authors have provided an overview of currently ongoing and published studies about stem cell therapy for myocardial infarction. Although we are dealing with a novel technology with limited applicability, it might be useful for readers’ understanding to provide the population size for each study (currently, it has been displayed only for reference n. 80).

Reviewer 2 Report

Comments and Suggestions for Authors

Dear Editors,

Thank you for the manuscript titled "Stem Cell Therapy for Myocardial Infarction Recovery: Advances, Challenges, and Future Directions" by N. Le et al., submitted to Biomedicines.

The authors have presented a comprehensive overview encompassing the current advancements in stem cell-based therapies for myocardial infarction. The manuscript effectively incorporates most of the recent developments and methodologies within this field. However, I have a couple of major concerns that I believe should be addressed to enhance the clarity and impact of the work:

1. The current version of Figure 1 appears overly simplistic and does not fully reflect the complexity or depth of the manuscript. I recommend revising it to include key information such as:

  • The various types of stem cells discussed (e.g., MSCs, iPSCs, ESCs)
  • Their sources of origin (e.g., bone marrow, adipose tissue, umbilical cord, etc.)
  • Modes of delivery (e.g., intramyocardial, intracoronary, intravenous)
  • Classification as autologous or allogeneic based on published clinical and preclinical data
    Incorporating these elements would provide readers with a clearer, more integrated visual summary of the therapeutic landscape presented in the manuscript.

2. The discussion on 3D bioprinting is currently limited and warrants a more detailed examination. Given the rapid progress and emerging importance of 3D bioprinting in cardiac regenerative medicine, I suggest elaborating on:

  • The types of bioinks and biomaterials used
  • The different bioprinting techniques (e.g., extrusion, inkjet, laser-assisted) and their applications in myocardial repair
  • The role of 3D bioprinting in creating vascularized, functional cardiac patches or tissue constructs
  • Relevant preclinical or clinical studies demonstrating its efficacy
    A more thorough discussion would provide valuable insights for readers and strengthen the manuscript's relevance in the context of cutting-edge regenerative strategies.

Thank you.

Round 2

Reviewer 2 Report

Comments and Suggestions for Authors

Dear Editor,

Thank you for the revised version of the manuscript. The authors have addressed all my concerns and I do not have any more comments.